# Learning-to-defer for sequential medical decision-making under uncertainty

**Shalmali Joshi**[†][*]                                                                    *shalmali.joshi@columbia.edu*
*Columbia University*

**Sonali Parbhoo**[*]                                                                        *s.parbhoo@imperial.ac.uk*
*Imperial College London*

**Finale Doshi-Velez**                                                                     *finale@seas.harvard.edu*
*Harvard University*

**Reviewed on OpenReview:** *https://openreview.net/forum?id=0pn3KnbH5F*

## Abstract

Learning-to-defer is a framework to automatically defer decision-making to a human expert when ML-based decisions are deemed unreliable. Existing learning-to-defer frameworks are not designed for sequential settings. That is, they defer at every instance independently, based on immediate predictions, while ignoring the potential long-term impact of these interventions. As a result, existing frameworks are myopic. Further, they do not defer adaptively, which is crucial when human interventions are costly. In this work, we propose Sequential Learning-to-Defer (SLTD), a framework for learning-to-defer to a domain expert in sequential decision-making settings. Contrary to existing literature, we pose the problem of learning-to-defer as model-based reinforcement learning (RL) to i) account for long-term consequences of ML-based actions using RL and ii) adaptively defer based on the dynamics (model-based). Our proposed framework determines whether to defer (at each time step) by quantifying whether a deferral now will improve the value compared to delaying deferral to the next time step. To quantify the improvement, we account for potential future deferrals. As a result, we learn a pre-emptive deferral policy (i.e. a policy that defers early if using the ML-based policy could worsen long-term outcomes). Our deferral policy is adaptive to the non-stationarity in the dynamics. We demonstrate that adaptive deferral via SLTD provides an improved trade-off between long-term outcomes and deferral frequency on synthetic, semi-synthetic, and real-world data with non-stationary dynamics. Finally, we interpret the deferral decision by decomposing the propagated (long-term) uncertainty around the outcome, to justify the deferral decision.

## 1 Introduction

Machine learning (ML) has the potential to be deployed for decision-making in complex domains such as healthcare, lending, and legal systems. In many cases, ML-based policy may not generalize to situations not encountered during training. In practice, it may be safer to defer to a human expert when using the ML policy may not improve outcomes or cause active harm. Automatically deferring to a human expert is called 'Learning-to-defer.' Earlier works have considered the problem of learning-to-defer in non-sequential settings (Mozannar and Sontag, 2020; Madras et al., 2017)..

In situations such as managing health, however, two key challenges remain. First, deferral decisions can significantly alter long-term outcomes. Thus modeling the long-term outcome is critical to decide *when* to

---

[*]Equal Contribution
[†]Corresponding Author

defer to an expert. Deferring too late may lead to unintended and irreversible harm. Deferring too early may increase the burden on the human expert. Second, when human interventions (after deferral) are costly, learning-to-defer *adaptively* and only when critical is crucial. To defer adaptively, we need a well-characterized model of the environment, a challenging estimation issue, especially under non-stationarity, i.e., when the dynamics of the environment change over time.

Existing learning-to-defer methods defer based on immediate outcomes e.g. Mozannar and Sontag (2020); Madras et al. (2017); Gennatas et al. (2020), and are therefore myopic. Further, the objective to defer is to improve the performance of some prediction tasks (such as the ability to predict a patient outcome). These frameworks either defer based on the probability of correct short-term prediction or characterizing the trade-off of paying a cost (to defer). Instead, interventions based on an ML system can have long-term consequences that are crucial to the model. Further, in many cases, merely deferring to optimize for decision/prediction accuracy in a supervised learning setting does not suffice to improve long-term outcomes. Existing approaches also do not leverage the potential of modeling the environment to defer *adaptively*, especially beneficial if the environment is non-stationary.

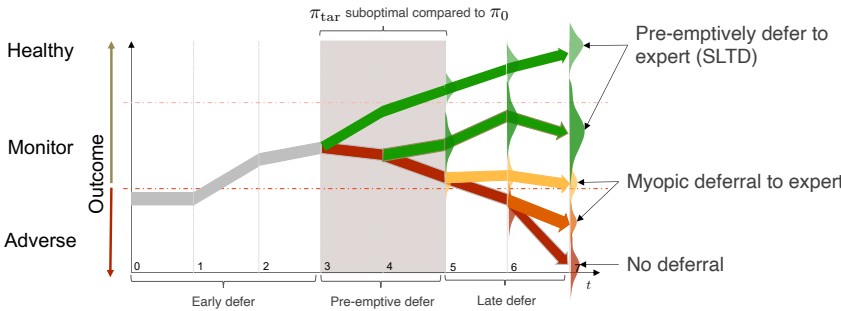

Figure 1: A conceptual overview of deferral strategies in a medical setting. We show potential patient outcome trajectories over time (x-axis). The outcome is shown on the y-axis. A higher outcome is better. A discretized reward is available to us: 'Healthy', 'Continue Monitoring', and 'Adverse'. The white regions indicate regions where $\pi_{\mathrm{tar}}$ will provide similar recommendations to $\pi_0$ (specific actions are not shown). The shaded gray region indicates that at these times, the patient is in states (not shown) where $\pi_{\mathrm{tar}}$ provides recommendations that do not improve long-term outcomes compared to $\pi_0$. The patient follows the gray trajectory from time $t = 0$ to $t = 2$ when $\pi_{\mathrm{tar}}$'s recommendations are used. The red trajectory shows the final outcome if we continue to use $\pi_{\mathrm{tar}}$ till the end of the horizon. One potential deferral strategy is to defer when the observed outcome deteriorates to 'Adverse' (orange trajectory). This deferral is late/myopic and the expert policy is unable to significantly improve the patient's outcome in a single time step. This baseline is akin to a model where the expert takes over from $\pi_{\mathrm{tar}}$ when they see that the patient is deteriorating and is not an automated deferral. A better-automated deferral strategy is to defer when the *predicted outcome* using $\pi_{\mathrm{tar}}$ is 'Adverse' (yellow trajectory). This deferral strategy corresponds to supervised learning-to-defer methods such as those of Mozannar et. al. and Madras et. al. and is also late/myopic as i) it is based on immediate predicted outcomes, and ii) does not characterize the regions of the state-space where $\pi_{\mathrm{tar}}$ is worse than $\pi_0$ (gray region) in terms of the long-term patient outcome. On the other hand, the green trajectories indicate long-term outcomes if we defer in the gray region (corresponding to what SLTD and SLTD-Stationary will do), where immediate predicted outcomes are not adverse, but long-term consequences of continuing to use $\pi_{\mathrm{tar}}$ significantly deteriorate the patient. Note that here we do not show the distinction between behaviors due to the non-stationarity of the dynamics. Thus, ideally, a pre-emptive deferral policy should characterize all regions of the state space where long-term outcomes using $\pi_{\mathrm{tar}}$ are worse than $\pi_0$, and defer accordingly. Also note that here, both green trajectories are pre-emptive according to our setup though SLTD will defer earlier at $t = 3$ as well as at $t = 4$ and then rely on $\pi_{\mathrm{tar}}$ for $t \geq 5$. In regions marked "Early defer", we would like to avoid deferrals since as $\pi_{\mathrm{tar}}$ is as good as or better than $\pi_0$ and deferring is unnecessarily costly. SLTD can avoid such premature deferrals by incorporating a cost for deferrals.

**Algorithmic Motivation.** To address these challenges, we deviate significantly from existing learning-to-defer methods, which use a supervised learning framework. Instead, we model the learning-to-defer problem for sequential settings as offline model-based reinforcement learning (RL). SLTD is the first RL-based learning-to-defer framework. We focus on settings where online experimentation is prohibitive for safety reasons, such as healthcare.

We assume access to batch data collected by human experts (such as clinicians) using a behavior policy. Our goal is to learn a deferral policy with respect to a fixed ML-based policy (called the target policy). SLTD decides whether or not to defer (to an expert policy) at each instance by modeling the impact of *delaying deferral* (by one time step) on the long-term outcomes. SLTD defers if delaying deferral does not improve outcomes compared to deferring in the current instance. To quantify long-term outcomes, we also account for all future deferrals. In doing so, SLTD precisely identifies the regions of the state space where the target ML-based policy will not improve outcomes. As a result, our method is pre-emptive, i.e., *it defers in all regions where the ML policy is unlikely to improve long-term outcomes*. See Figure 1 for a conceptual overview of SLTD.

Human expert interventions are often costly. Hence deferring too often is not desirable. To defer *adaptively*, we propose to leverage an estimate of the environment dynamics and the associated uncertainty. Modeling the dynamics allows us to reliably quantify the impact of delaying deferral on long-term outcomes, which is particularly beneficial in non-stationary settings. We show that modeling the non-stationarity provides a better trade-off of improving outcomes versus the frequency of deferrals. In a myopic environment (i.e., when the effect of interventions are observed in the near future), it may seem unnecessary to model the dynamics. However, we demonstrate that deferral methods that defer myopically, based on immediate outcomes still benefit from modeling the dynamics, and consequently the impact of potential future myopic deferrals. When SLTD defers, human experts can benefit from an additional justification of the deferral decision to determine potential interventions. Hence, we also interpret SLTD's decision to defer at any given time by quantifying the long-term uncertainty in the outcome and decomposing the sources of uncertainty. We justify how the decomposition can guide experts to potential interventions.

**Clinical Motivation.** We are motivated by clinical settings where a target policy is learned from batch data to work well across multiple institutions. Such a policy may perform well on average, but when deployed to a new environment, encounter a different or an evolving patient population. An evolving patient physiology can often result in non-stationary dynamics on which the target policy is not uniformly better, necessitating deferral. Clinicians might also follow a slightly different treatment protocol than this learned policy, which could be an auxiliary reason to defer. Most importantly, regulatory constraints may prevent significantly adapting our target policy completely to the new site. In this case, it is safer to leverage batch data from the new site to quantify when it is reliable to deploy the learned target policy. In situations where the policy does not improve outcomes compared to the human expert, exacerbated by challenges like non-stationarity, it is safer to defer to human experts. We further propose a model-based method motivated by the fact that in many clinical settings, mechanistic models of patient physiology are available. Beyond healthcare, our work is applicable in many safety-focused, data-scarce, non-stationary settings where online policy improvement is not allowed due to ethical or practical constraints.

## 2 Related Work & Background

**Mixture-of-Experts (MoE).** Many methods focus on deciding to deploy two or more policies. For example, Jacobs et al. (1991); Jordan and Jacobs (1994) switch between different policies in decision-making by partitioning the input space into regions assigned to different specialized sub-models. Variants of this framework enforce an explicit preference for a specific expert, e.g., a human expert, and train other experts to complement the human expert (Pradier et al., 2021). In sequential settings, Parbhoo et al. (2017); Gottesman et al. (2019); Parbhoo et al. (2018) combine parametric and non-parametric experts to learn more accurate estimates of the value function. On the other hand, we focus on *deferral to human experts when future outcomes using the current ML-based policy are potentially undesirable*. Further, we defer based on explicitly quantifying the impact of delayed deferral to decide *when* to defer.

**Policy Improvement with Expert Supervision.** Sonabend et al. (2020) use hypothesis testing to assess whether, at each state, a policy from a human expert would improve value estimates over a target policy

*during training* to improve the target policy. In contrast, our work identifies the value of *delaying deferral to a human expert at test time.* Improvements using expert supervision are unlikely to be always feasible due to safety and regulatory constraints. Learning-to-defer with respect to a fixed target policy is crucial as a safeguard. Some works focus on safe policy improvement in a non-stationary MDP setting (Chandak et al., 2020b;a). Chandak et al. (2020a) assume that the non-stationarity is governed by an exogenous process, and so past actions do not impact the underlying non-stationarity. Our work differs in two ways: first, we argue that model misspecification, specifically ignoring non-stationarity induced by (deferral) actions, affects the likelihood of future deferrals. Accounting for this non-stationarity is crucial to avoid costly deferrals. Second, we incorporate human expertise by explicitly measuring the impact of delaying deferral.

**Learning-to-defer to Human Expertise.** Madras et al. (2017); Mozannar and Sontag (2020) propose supervised models to defer to the expert. Here, the classifiers are trained on the samples of an expert's decisions. Madras et al. (2017) train a separate rejection and prediction function, while Mozannar and Sontag (2020) learn a joint predictor for all targets and deferral. Madras et al. (2017) is conceptually closer to our work but in a non-sequential setting. Other approaches such as Raghu et al. (2019); Wilder et al. (2020) first train a standard classifier on the data and then compute uncertainty estimates for this classifier and the human expert. The models defer to the expert if the model is highly uncertain or can significantly benefit from deferral. Liu et al. (2021) incorporate uncertainty in Learning-to-Defer algorithms for classification tasks. Instead, we focus on learning-to-defer in non-stationary, sequential settings. Our work highlights the role that the dynamics of the environment can have on our ability to defer preemptively.

**Learning-to-defer as Causal OPE.** We pose our Learning-to-defer problem as an offline model-based reinforcement learning problem. Besides learning, evaluating the utility of a policy (on data collected from a behavior policy) in an offline manner is called Off-policy Evaluation (OPE) (Precup, 2000). OPE is a challenging problem as the utility of a policy needs to be determined without exploration. Literature on OPE is extensive, summarized in a seminal review of (Uehara et al., 2022). In model-based settings, Importance Sampling (IS), and its variants, and a Direct Method (DM) that directly estimates the Q-function, are the most foundational methods for OPE. Offline learning refers to learning an improved policy in offline settings. Q-function estimation requires making parametric assumptions of the Action-value functions, while highly flexible parametrizations are prone to overfitting that may not generalize. We strongly believe this is less grounded for healthcare settings. On the other hand, mechanistic models of disease and physiology are often available in clinical settings, motivating our model-based approach to offline learning. For evaluating the quality of our proposed method, we rely on either the knowledge of the true dynamics or IS in this work. IS is asymptotically unbiased but can suffer from variance challenges in finite-sample settings. For real-world data, we use the self-normalized variant of importance sampling from Uehara et al. (2022); Precup (2000); Robins et al. (2007) as one of the evaluation metrics. Nonetheless, overlap assumptions are critical to obtaining unbiased OPE estimates using IS, a crucial assumption that may be violated in practice, especially under non-stationarity. Quantifying overlap in a data-driven manner is challenging (Oberst et al., 2020). in this work, we assume sufficient overlap between clinician/expert policy and behavior policy during evaluation.

Implicit in the framework of OPE, including our work, are assumptions about no hidden confounding, which are surfaced by a causal framing of the OPE problem (Uehara et al., 2022; Gottesman et al., 2018). This view poses OPE as a causal inference problem given observations from a causal system. The no-hidden confounding assumption is primarily because most OPE solutions assume that the data is generated from an MDP which does not allow for potential latent factors to drive decisions. This assumption can have profound consequences on the quality of OPE estimates. In our setup, we also assume that our offline data is generated from an MDP, thus making the no unobserved/hidden confounding assumption.

Efforts to relax this assumption allow for the presence of latent factors that influence decision-making in off-line data and provide OPE estimates that are robust to the variability of this influence (Kallus and Zhou, 2018; Tennenholtz et al., 2020; Oberst and Sontag, 2019). Often these works focus on parametric assumptions of how much the propensity or the probability of a particular treatment is allowed to deviate, known as the Marginal Structural Model (MSM) assumption (Robins et al., 2000) to provide conservative OPE estimates under worst-case deviations under the MSM assumption. The utility of MSM assumptions

has been significant in epidemiological settings, though relevance in chronic condition management, which is the focus of our clinical setting, is less clear.

**Decomposing Uncertainty for Interpreting Policies.** Uncertainty, if well calibrated can help decision-makers understand the failure modes of a model (Bhatt et al., 2020; Tomsett et al., 2020; Zhang et al., 2020). Several methods estimate predictive uncertainty in ML (Gal and Ghahramani, 2016; Guo et al., 2017). Here, we focus on capturing the *propagated uncertainty* in sequential settings to interpret deferral decisions. We interpret the (different) sources of propagated uncertainty when SLTD defers to the expert. Decomposing the sources of uncertainty into modeling and irreducible uncertainty over predictions has been explored in classification and prediction settings (Yao et al., 2019; Depeweg et al., 2018) but remains significantly under-explored for sequential settings.

**Background and Notation.** We consider our environment to be a finite horizon MDP defined by $\mathcal{M} \equiv (\mathcal{S}, \mathcal{A}, \mathcal{P}, r, p_0)$ where $\mathcal{S}$ indicates the state-space, $\mathcal{A}$ indicates the action-space, $\mathcal{P}$ the transition dynamics, $r : s \times a \to \mathbb{R}_+$ the reward function, $p_0$ the initial state distribution. The action-space is assumed to be discrete, while the state space can be discrete or continuous. Any intervention policy (usually stochastic in our case) is given by $\pi =: \mathcal{S} \times \mathcal{A} \to [0, 1]$. We consider a non-stationary environment such that the dynamics at any time $t$ are governed by a specific MDP $\mathcal{M}_t \equiv \mathcal{P}_t(s'|s, a)$. Thus the environment is a sequence of MDPs. We assume the existence of a true set of non-stationary dynamics governing all episodes and denote it by $\mathcal{M}^* := \{\mathcal{M}_t^*\}_t$. In the rest of the draft, $\mathcal{M} := \{\mathcal{M}_t\}_t$ denotes an estimate of the true dynamics $\mathcal{M}^*$. Let $T$ be the episode-length. The value of a policy $\pi$ at $t$ is given by $V_{\pi,t}^{\mathcal{M}}(s) = \mathbb{E}_{\mathcal{M}}[\sum_{j=t}^{T} r^j(s, a)|s_t = s, \pi]$. The action value is given by $Q_{\pi,t}^{\mathcal{M}}(s, a) = r(s, a) + \sum_{s' \in \mathcal{S}} \mathcal{P}_t(s'|s, a) V_{\pi,t}^{\mathcal{M}}(s')$.

## 3 Sequential Learning-to-Defer

**Problem Setup.** Assume we are given a policy $\pi_{\text{tar}}$ that may be learned from batch data from one or more environments. $\pi_{\text{tar}}$ is intended to be deployed in a new environment. We have access to batch data, denoted by $\mathcal{D}^* = \{s_{i,0}, a_{i,0}, r_{i,0}, \cdots, s_{i,T}, a_{i,T}, r_{i,T}\}_{i=1}^{N}$ collected in the new non-stationary environment $\mathcal{M}^* = \{\mathcal{M}_t^*\}_t$, from some (potentially non-stationary) expert policy $\pi_0$. Note that we assume that $\pi_0$ is given. For example, it may be the behavior policy from which we have data samples in the target environment. Here $N$ denotes the number of episodes. Our goal is to learn a deferral policy $g_{\pi_{\text{tar}}}(s, t) : \mathcal{S} \times T \to \{0, 1\}$ (where 1 corresponds to defer or $\perp$) with respect to $\pi_{\text{tar}}$ to defer to the expert policy $\pi_0$. In practice, experts may deviate from $\pi_0$ in some cases. In our experiments, we account for this by using an $\epsilon$-greedy version of the behavior policy as $\pi_0$, which serves as a proxy model for such deviation. In addition, a clinician may override a treatment recommendation even when a model does not defer. We account for this using $\epsilon$-greedy version of $\pi_{\text{tar}}$. For ease of exposition, we still refer to them as $\pi_0$ and $\pi_{\text{tar}}$.

Deferral to the expert is denoted by the action $\perp$. That is, we will augment the action space of existing MDP $\mathcal{M}^*$ to include a new deferral action $\mathcal{A}_\perp := \mathcal{A} \cup \perp$. At every step, the agent decides whether or not to defer. If the agent defers, $\pi_0$ will be deployed for that time step. We describe the formulation assuming strict adherence to $\pi_0$ at deferral to emphasize other aspects of our contribution such as the impact of non-stationarity and how to account for relevant sources of uncertainty to compare outcomes. SLTD can easily account for the uncertainty of expert actions in the framework.

In practice, the target policy $\pi_{\text{tar}}$ may not uniformly improve over $\pi_0$ for all states. That is guaranteeing that $V_{\pi_{\text{tar}},0}^{\mathcal{M}^*}(s) \geq V_{\pi_0,0}^{\mathcal{M}^*}(s)$ for all $s \in \mathcal{S}$, is challenging. Even when fine-tuning is allowed, it is challenging to ensure that the target policy is indeed better than $\pi_0$ in all regions of the state space. Hence, we would like to get the best of both worlds. We can deploy $\pi_{\text{tar}}$, to reduce the costs of relying on human expertise, and learn to automatically defer to the costlier policy $\pi_0$ (i.e. human expert) when relying on $\pi_{\text{tar}}$ does not improve outcomes. In regions of the state-space where the value of $\pi_{\text{tar}}$ is lower than $\pi_0$, it is better to defer to the human as a "safety protocol". Formally, we only assume that $V_{\pi_{\text{tar}}}(s) > V_{\pi_0}(s)$ for some states $s \in \mathbf{s}$.

Type of non-stationarity: Assuming that the non-stationary dynamics are represented by a sequence of MDPs allows the SLTD framework to be general and not restricted to specific forms of non-stationary environments. The main difference between each component in the MDP sequence is that they can be arbitrarily different state-transition dynamics within the same family of distributions (e.g. gaussian or multinomial distributions). As a result, this sequence will not share the optimal policy, and hence the optimal is a non-stationary

deterministic policy. When mechanistic models on the specific *type* of non-stationarity are available, they can be incorporated into our framework to provide less conservative deferral policies.

**SLTD.** To determine whether to defer at each time step, we quantify whether deferring (relying on $\pi_0$) or not deferring (using $\pi_{\text{tar}}$) at the current time step improves the long-term outcome. Long-term outcomes are affected by potential future deferrals. Thus comparing the consequences of deferring versus relying on the ML policy at the current instance is equivalent to comparing the impact of deferring now versus delaying deferral by one time step.

Future deferrals imply that some unknown mixture of $\pi_{\text{tar}}$ and $\pi_0$ is used in the future. We denote such a mixture policy as $\pi_{\text{mix}}$. To minimize cumbersome notation, we denote a policy where $\pi_{\text{tar}}$ is deployed at instance $t$ and $\pi_{\text{mix}}$ in the future as: $\pi_{\text{tar}(t),\text{mix}(t_+)}$. Similarly, if we defer *now*, then the policy that is deployed at time $t$ is $\pi_0$, and $\pi_{\text{mix}}$ in the future. We denote this mixture as $\pi_{0(t),\text{mix}(t_+)}$. Thus, at any instance $t$, we want to defer if $V^{\boldsymbol{\mathcal{M}}}_{\pi_{\text{tar}(t),\text{mix}(t_+)}}(s) < V^{\boldsymbol{\mathcal{M}}}_{\pi_{0(t),\text{mix}(t_+)}}(s)$. Note that we consider deferral to $\pi_0$ as a costly one. This is accounted through a constant cost $c > 0$ in terms of the value. That is, deferral incurs cost $c$ and the resulting value is: $V^{\boldsymbol{\mathcal{M}}}_{\pi_{0(t),\text{mix}(t_+)}}(s) - c$. We can now formalize our stochastic deferral policy:

**Definition 1.** *Let $\pi_{tar,t}$ be such that there exists $s^t \subseteq \mathcal{S} \; \forall t \in \{0, 1, \cdots, T\}$ where $P(V^{\boldsymbol{\mathcal{M}}}_{\pi_{tar(t),mix(t_+)}}(s) < V^{\boldsymbol{\mathcal{M}}}_{\pi_{0(t),mix(t_+)}}(s) - c) > \tau$ for constant cost of deferral $c > 0$ and threshold $\tau > 0$, $\forall s \in s^t$. Then the deferral policy $g_{\pi_{tar}}(s, t) \triangleq \mathbf{1}[P(V^{\boldsymbol{\mathcal{M}}}_{\pi_{tar(t),mix(t_+)}}(s) < V^{\boldsymbol{\mathcal{M}}}_{\pi_{0(t),mix(t_+)}}(s) - c) > \tau] \triangleq \mathbf{1}[\tilde{g}_{\pi_{tar}}(s, t) > \tau]$.*

**Corollary 1.** *By Definition 1, $g_{\pi_{tar}}(s, t)$, includes the earliest time in the episode where $\tilde{g}_{\pi_{tar}}(s, t) \triangleq P(V^{\boldsymbol{\mathcal{M}}}_{\pi_{tar(t),mix(t_+)}} < V^{\boldsymbol{\mathcal{M}}}_{\pi_{0(t),mix(t_+)}} - c) > \tau$. Thus, $g_{\pi_{tar}}(s, t)$ is a pre-emptive deferral policy.*

The cost $c$ determines how conservative SLTD is and trades-off frequency of deferral to the value attained. This parameter should be tuned by domain experts aware of the trade-off and risks involved. For instance, in a critical care setting, we may be more conservative and use a smaller $c$ than in a chronic care situation. $\tau$ is a safety threshold on the probability of worse outcome beyond which we deem that deferral is necessary.

Definition 1 indicates that to reliably learn the deferral policy, we need to estimate $\tilde{g}_{\pi_{\text{tar}}}(s, t) \triangleq P(V^{\boldsymbol{\mathcal{M}}}_{\pi_{\text{tar}(t),mix(t_+)}}(s) < V^{\boldsymbol{\mathcal{M}}}_{\pi_{0(t),mix(t_+)}}(s) - c)$. To estimate this probability, we should model all sources of uncertainty in the system, including the non-stationary dynamics, and the uncertainty associated with our modeling assumptions. We use a Bayesian RL approach to account for all sources of uncertainty. We motivate this by first describing our dynamic programming approach to learn-to-defer.

Our dynamic programming procedure maintains an estimate of the deferral probability $\tilde{g}_{\pi_{\text{tar}}}(s, t)$ and refines it as we train on the batch data. Given an estimate of $\tilde{g}_{\pi_{\text{tar}}}(s, t)$, we outline the procedure to i) estimate the value under mixture policies corresponding to deferral (and delayed deferral), ii) modeling the probability of improvement under various sources of uncertainty, and finally iii) obtaining a new estimate of the deferral probability $\tilde{g}_{\pi_{\text{tar}}}(s, t) \forall s \in \mathcal{S}$ at the given time $t$ using i) and ii). We then bootstrap this procedure over our batch data to refine our deferral probabilities. We describe the procedure for estimating the dynamics in the discrete setting.

**Estimating Value function.** At any instance we defer based on current estimates of $g_{\pi_{\text{tar}}}(s, t)$ (or equivalently $\tilde{g}_{\pi_{\text{tar}}}(s, t)$). We sample actions from $\pi_{\text{tar}}$ if $g_{\pi_{\text{tar}}}(s, t) = 0$ and $\pi_0$ otherwise (equivalent to $\perp$). Note that the current estimate of $g_{\pi_{\text{tar}}}(s, t)$ determines the future mixture policy as well. We now estimate the value of the mixture policies using the Bellman Equation of the state and action value functions. For the mixture policy $\pi_m \triangleq \pi_{tar(t),mix(t_+)}$ (corresponding to no deferral at $t$), the Q-function is:

$$Q^{\boldsymbol{\mathcal{M}}}_{\pi_m,t}(s, a) = r(s, a) + \sum_{s' \in \mathcal{S}} \mathcal{P}^{\boldsymbol{\mathcal{M}}}(s'|s, a) V^{\boldsymbol{\mathcal{M}}}_{\pi_{\text{mix}(t_+)},t+1}(s') \tag{1}$$

and the Value function is:

$$V^{\boldsymbol{\mathcal{M}}}_{\pi_m,t}(s) = \sum_{a \in \mathcal{A}} \pi_{\text{tar}(t)}(a|s) Q^{\boldsymbol{\mathcal{M}}}_{\pi_m,t}(s, a) \tag{2}$$

Similarly for the mixture policy if we defer at $t$.

---

**Algorithm 1** Sequential Learning to Defer

---

**Input:** $\mathcal{D}^*$, expert policy $\pi_0$, target policy $\pi_{\text{tar}}$.
Estimate Posterior Distributions $\{\mathcal{M}_t \triangleq p_t(\cdot|\mathcal{D}^*)\}_{t=0}^T$ (posteriors over rewards not shown here)
**Initialization:** Deferral function $g_{\pi_{\text{tar}}}(s,t) = 0$ for all $s \in \mathcal{S}$ and $t \in \{1, 2, \cdots, T\}$.
**for** $n \in \text{BOOTSTRAPS}(\mathcal{D}^*)$ **do**
    Sample $\boldsymbol{\mathcal{M}}_k \triangleq \{\mathcal{M}_{k,t} \sim p_t(|\mathcal{D}*)\} \forall t \in \{1, 2, \cdots, T\}, \forall k \in \{1, 2, \cdots, K\}$
    **for** $t \in \{T, T-1, \cdots, 1\}$ **do**
        **for** $s \in \mathcal{S}$ **do**
            Compute $V_{\pi_{\text{tar}(t)},\text{mix}(t_+)}^{\boldsymbol{\mathcal{M}}}, V_{\pi_{0(t)},\text{mix}(t_+)}^{\boldsymbol{\mathcal{M}}} - c \, \forall \boldsymbol{\mathcal{M}}$
            $\tilde{g}_{\pi_{\text{tar}}}(s,t) \leftarrow \approx \frac{1}{K} \sum_{\boldsymbol{\mathcal{M}}_k \sim \{p_{t'}(\cdot|\mathcal{D})\}_{t'=t}^T} [\mathbf{1}(V_{\pi_{\text{tar}(t)},\text{mix}(t_+)}^{\boldsymbol{\mathcal{M}}_k} < V_{\pi_{0(t)},\text{mix}(t_+)}^{\boldsymbol{\mathcal{M}}_k} - c)]$
        **end for**
    **end for**
**end for**
**return** $g_{\pi_{\text{tar}}}(s,t) = \mathbf{1}(\tilde{g}_{\pi_{\text{tar}}}(s,t) > \tau) \forall s, t \in \mathcal{S} \times \{1, 2, \cdots, T\}$

---

**Estimating the probability of improving outcomes by delaying deferral.** At each instance $t$, for all states $s$, we can estimate the indicator function $\mathbf{1}[V_{\pi_{\text{tar}(t)},\text{mix}(t_+)}^{\boldsymbol{\mathcal{M}}^*}(s) < V_{\pi_{0(t)},\text{mix}(t_+)}^{\boldsymbol{\mathcal{M}}^*}(s) - c]$ given an estimate of $\tilde{g}_{\pi_{\text{tar}}}$ as described above. However, we do not have access to the true dynamics $\boldsymbol{\mathcal{M}}^*$. In batch settings, such as ours, we often estimate the dynamics using maximum-likelihood estimation. Such methods make specific assumptions about the distribution governing the dynamics. Our assumptions about the dynamics may be incorrect resulting in potential misspecification of our dynamics model. This increases the uncertainty in the outcome and potentially over-estimates the probability that relying on the model may improve outcomes. To account for this additional source of uncertainty, we use a Bayesian RL approach. We describe the procedure for the dynamics. The procedure for rewards follows an analogous process.

Suppose the parameters of the distributions governing the dynamics are denoted by $\theta_t \, \forall t \in \{0, \cdots, T\}$. We denote the full set of parameters by $\boldsymbol{\theta} = \{\theta_t\}_t$. We assume a prior distribution over the parameters of the distribution governing the dynamics $\mathcal{P}_\theta^{\boldsymbol{\mathcal{M}}}(s'|s,a)$ and the rewards $r(s,a)$. Given batch samples $\mathcal{D}^*$, we can estimate the posterior distribution over the non-stationary MDPs and rewards using Bayesian inference:

$$p(\boldsymbol{\theta}|\mathcal{D}^*) \propto p(\mathcal{D}^*|\boldsymbol{\theta})p(\boldsymbol{\theta})$$

More specifically, we assume conjugate priors for our parameters $\boldsymbol{\theta}$. By relying on conjugate priors in our inference, the parameters of posterior distributions over the dynamics and rewards are obtained in closed form. For discrete state dynamics (and rewards), we assume a Dirichlet prior distribution and model the observations $p(\mathcal{D}^*|\boldsymbol{\theta})$ using a Multinomial distribution. For continuous states, $p(\mathcal{D}^*|\boldsymbol{\theta})$ is assumed to be normally distributed with $\boldsymbol{\theta}$ being the mean and variance parameters. The prior distributions over the mean and precision (inverse of the variance) is the Normal-gamma prior. This is a domain-dependent choice and SLTD is agnostic so long as we can sample from the posterior distributions of the learned model dynamics. A detailed derivation of how the data is leveraged to estimate the posterior distributions over the dynamics are provided in Appendix A.1. By allowing flexibility of modeling the dynamics via Bayesian RL, we can account for uncertainty over our modeling assumptions.

Finally, based on our assumption that the non-stationary environment is governed by a sequence of MDPs, we estimate the MDP for each time step independently from batch data. This allows us to make fewer assumptions about the *type* of non-stationarity. Any additional domain knowledge about the nature of non-stationarity can be leveraged for data efficiency. We can now estimate the impact of delayed deferral by sampling non-stationary MDPs from our posterior distributions and averaging to obtain our final probability:

$$
\begin{aligned}
\tilde{g}_{\pi_{\text{tar}}}(s,t) &\triangleq P(V_{\pi_{\text{tar}(t)},\text{mix}(t_+)}^{\boldsymbol{\mathcal{M}}^*}(s) < V_{\pi_{0(t)},\text{mix}(t_+)}^{\boldsymbol{\mathcal{M}}^*}(s) - c) \\
&= \mathbb{E}_{\boldsymbol{\mathcal{M}} \sim p(\cdot|\mathcal{D}^*)}[\mathbf{1}[V_{\pi_{\text{tar}(t)},\text{mix}(t_+)}^{\boldsymbol{\mathcal{M}}}(s) < V_{\pi_{0(t)},\text{mix}(t_+)}^{\boldsymbol{\mathcal{M}}}(s) - c]] \\
&\approx \frac{1}{K} \sum_{\boldsymbol{\mathcal{M}}_k \sim \{p_{t'}(\cdot|\mathcal{D}^*)\}_{t'=t}^T} \mathbf{1}[V_{\pi_{\text{tar}(t)},\text{mix}(t_+)}^{\boldsymbol{\mathcal{M}}_k}(s) < V_{\pi_{0(t)},\text{mix}(t_+)}^{\boldsymbol{\mathcal{M}}_k}(s) - c]
\end{aligned}
\tag{3}
$$

where the second line comes from the definition due to the randomness over the dynamics, and the last term comes from approximating the expectation using $K$ samples from the posterior distribution of the dynamics $p(\cdot|\mathcal{D}^*)$. Thus, for every instant $t$, in a given state $s$, our deferral policy $g_{\pi_{\text{tar}}}(s,t)$ is given by, $g_{\pi_{\text{tar}}}(s,t) := \mathbf{1}[\tilde{g}_{\pi_{\text{tar}}}(s,t) > \tau]$.

**Dynamic Programming to estimate** $g_{\pi_{\text{tar}}}(s,t)$. Our dynamic programming procedure is summarized in Algorithm 1. We initialize $g_{\pi_{\text{tar}}}(s,t) = 0$ for all $s \in \mathcal{S}$. We estimate $V^{\mathcal{M}}_{\pi_{\text{tar}(t)},\text{mix}(t_+)}(s), V^{\mathcal{M}}_{\pi_{0(t)},\text{mix}(t_+)}(s)$ for a given $t$ using Bellman Equations 1 and 2. Following that, we can update our estimate of $g_{\pi_{\text{tar}}}(s,t)$ using our posterior MDPs, i.e. Equation 3. We repeat (over $t$) using the *updated* estimates of $g_{\pi_{\text{tar}}}(s,t)$. Note further that $\pi_0$ is stochastic. Thus, we do not make explicit assumptions on the specific actions an expert will take in the futher. More specifically, SLTD accounts for the added uncertainty in human's actions by explicitly taking expectations over actions $a \sim \pi_0$. In our experiments, we use an $\epsilon$-greedy versions of both $\pi_0$ and $\pi_{\text{tar}}$, which will allow for further deviations from the expert policy or allowing for overrides, to reflect realistic settings.

**Optimality of Learned Deferral Policy.** Note that the optimal policy in our environment is a deterministic non-stationary policy. More specifically, the optimal policy is a sequence of policies where each component in the sequence is optimal with respect to the specific dynamics at the corresponding time instance. It may not be possible to always reach such an optimal via deferral. Our goal is thus to significantly improve over our target policy $\pi_{\text{tar}}$ by deferring to the expert policy. This could be envisioned as a setup where are restricted to a policy class of offline RL where we are only allowed to learn from families that defer to the expert policy and use $\pi_{\text{tar}}$ otherwise.

## 4  Decomposing the uncertainty at deferral

SLTD defers at time $t$ because the probability that relying on $\pi_{\text{tar}}$ improves the outcome is below our safety threshold, i.e. SLTD is uncertain of an improved outcome. Conveying this uncertainty can help the domain expert take over decision-making. We interpret this deferral decision in terms of the total and decomposed uncertainty on long-term outcomes. We convey two different sources of uncertainty at deferral. First, we consider *epistemic/modeling uncertainty*, which captures whether our model specification has resulted in high uncertainty and the *aleatoric uncertainty* which mainly results from the stochasticity of the environment itself. A high relative value of the former suggests that adding more data to train SLTD can improve the confidence of the model. High aleatoric uncertainty suggests that the environment itself is highly variable leading to the lack of confidence in relying on $\pi_{\text{tar}}$.

Concretely, let $t_d$ be a time when SLTD defers. The agent is in state $s_{t_d}$. We are interested in the reward (and uncertainty over the reward) at time $T$ due to deferral at $t_d$, i.e., $\mathbb{E}[r_T|s_{t_d}, \mu_{t_d}, \pi_{0(t_d),mix(t_d+)}]$. We denote the posterior MDP samples for any state-action pair by $\mu_t$. The variability in these samples captures modeling uncertainty. The dynamics parameters are denoted by $\theta_t(s,a)$ for each state-action pair. First, we sample the parameters of the dynamics from posterior distribution $p(\theta_{t'}|\mathcal{D}^*)$, followed by sampling the MDPs $\mu_{t'} \sim p(\mu_{t'}|\theta_t'(s_{t'},a_{t'}))$. Once we defer, we sample actions from $\pi_0$ at time $t' = t_d$ and $\pi_{\text{mix}}$ for $t' > t_d$ where the mixture probability is determined by the learned $g_{\pi_{\text{tar}}}$ for future deferrals. The expected long-term outcome is given by:

$$\mathbb{E}[r_T|s_{t_d}, \mu_{t_d}] = \int_{s_{t_d+1}}^{s_T} \int_{a_{t_d}}^{a_T} \int_{\mu_{t_d+1}}^{\mu_T} \int_{\theta_{t_d}}^{T} r(s_T, a_T) \times \prod_{t'=t_d+1}^{T} p_{t'}(s_{t'}|\mu_{t'})p_{t'}(\mu_{t'}|\theta_t'(s_{t'},a_{t'}))\pi_{t'}(a_{t'}|s_{t'})p_{t'}(\theta_{t'}|\mathcal{D})d\mathbf{s}d\mathbf{a}d\boldsymbol{\mu}d\boldsymbol{\theta}$$

Integrands are written in short-hand: $\mathbf{s} = \{s_{t_d+1}, s_{t_d+2}, \cdots, s_T\}$ (analogously for other quantities). We maintain one estimate of parameter $\theta_{t'}$ and sample $K$ MDPs $\mu_{t'}$ from this distribution. Thus, the epistemic uncertainty we capture is due to the uncertainty over dynamics under fixed parameters. The total uncertainty can now be decomposed using the law of total variance:

$$\underbrace{\text{Var}(r_T|s_{t_d}, \mathcal{D})}_{\text{Total Uncertainty}} = \underbrace{\mathbb{E}_{\mu_{t_d} \sim p(\mu_{t_d}|\mathcal{D})}[\text{Var}(r_T|\mu_{t_d}, s_{t_d}, \mathcal{D})]}_{\text{Irreducible/ Aleatoric Uncertainty}} + \underbrace{\text{Var}_{\mu_{t_d} \sim p(\mu_{t_d}|\mathcal{D})}(\mathbb{E}[r_T|\mu_{t_d}, s_{t_d}, \mathcal{D}])}_{\text{Epistemic/Modeling Uncertainty}}$$

The second term is the variance *conditioned* on knowledge of the model $\mu_{t_d}$. This is the *propagated uncertainty due to modeling uncertainty at $t_d$* and can be reduced by data collection. The first term averages over the

variance due to $\mu_{t_d}$ and captures *propagated uncertainty due to aleatoric uncertainty at $t_d$*, which conveys stochasticity of the environment itself. This uncertainty can only be reduced by careful interventions at $t_d$. We estimate these using Monte-Carlo sampling. Additional details on the derivation are provided in Appendix A.2. As suggested before, a high *propagated epistemic uncertainty* conveys that the current uncertainty of model prediction (of the dynamics) is high but could be improved if additional data could be collected. High *propagated aleatoric uncertainty* indicates high variability in the dynamics that can only be reduced with careful interventions and is otherwise not manageable.

## 5 Experiments

We evaluate SLTD's ability to defer adaptively in sequential settings with respect to a known and fixed $\pi_{\text{tar}}$ to the expert policy $\pi_0$. We test the utility of i) deferring based on long-term outcomes, ii) adaptively deferring by quantifying the impact of delaying deferral, i.e., in regions where delayed deferral can worsen outcomes, iii) modeling the non-stationarity on deferral frequency, iv) quantifying multiple sources of uncertainty to estimate the probability of different outcomes under delayed deferral. We test our method on synthetic data, a non-stationary diabetes simulator modified from Chandak et al. (2020b), and real-world HIV data.

**Synthetic Data.** In this synthetic simulation, the region of deferral is known apriori by careful design of $\pi_{\text{tar}}$. This environment has 8 discrete states and binary actions $\{a_0, a_1\}$. All samples start at state 0 and progress toward a sink state 7. The episode length is 15. State 6 has a low reward $(-5)$ while all other states have a reward of $+1$. The initial dynamics are set up such that action $a_0$ reduces the probability of landing in stage 6, and action $a_1$ increases the probability of reaching state 6. $\pi_{\text{tar}}$ increases the chances to reach state 6 unfavourably by taking action $a_1$ in states $2, 3, 4$ when $t < 5$ or $t > 12$. We expect to defer in states $2, 3, 4$ even though rewards are favorable if a method is pre-emptive. When $5 \leq t \leq 12$, the dynamics flip such that $a_0$ becomes an unfavorable action that increases the probability of landing in 6, while $a_1$ reduces this probability. Here, $\pi_{\text{tar}}$ again increases the chances of landing in 6, by taking $a_0$ more often in states $2, 3, 4$. By flipping the better action to $a_0$ in this region, it becomes crucial to *estimate the dynamics* over predicting the best action. The dynamics are non-stationary and the probability of landing in state 6 progressively increases when $5 \leq t \leq 12$. For $t \geq 13$, the dynamics reset to noise levels at $t < 5$ adding non-stationarity to the dynamics. Note that the optimal policy is $\pi(s, t) = 1 \forall s \in \mathcal{S}, 5 \leq t \leq 12$ and 0 otherwise. The expert policy is such that:

$$\pi_0(s, t) := p(a_1) = \begin{cases} 0.9 & \text{if } 5 \leq t \leq 12, s \in \{2, 3, 4\} \\ 0.7 & \text{if } 5 \leq t \leq 12, s \notin \{2, 3, 4\} \\ 0.1 & t < 5 \text{ or } t > 12, \forall s \in \mathcal{S} \end{cases}$$

for all states $s \in \mathcal{S}$. In this case, pre-emptive deferral will allow us to reach close to the optimal by deferring in states $\{2, 3, 4\}$.

**Real-world simulator: Diabetes Data.** We use an open-source implementation of the FDA-approved Type-1 Diabetes Mellitus simulator (T1DMS) for modeling the treatment of Type-1 diabetes. We sample 10 adolescent patient trajectories (episodes) over 24 hours (aggregated at 15 minute intervals). Glucose levels are discretized into 13 states. Combination interventions of insulin and bolus are discretized to generate a total of 25 actions. We introduce non-stationarity in each episode by increasingly changing the adolescent patient's properties to an alternative patient. We enable this by smoothly varying the weighting of the patient parameters over the horizon. While this does not reflect a realistic patient scenario but will nonetheless evaluate the utility of all methods for a smoothly transitioning non-stationary environment. The non-stationarity significantly affects the utility of the initial target policy which is learned on the dynamics of the original patient, thus necessitating deferral as the patient properties change over time. The non-stationary target policy $\pi_{\text{tar}}$ for this task is estimated using Q-learning. An $\epsilon$-greedy version of this Q-learned policy is used in our experiments. We defer to a clinician policy, here simulated by learning an $\epsilon$-greedy version of a policy learned using Q-learning under (estimated) non-stationary dynamics on the target data. For evaluation of value post learning, we estimate the dynamics on $N = 1000$ patients to remove estimation bias for evaluation purposes. We further provide IS estimates as we discuss in the metrics below.

Jinyu Xie. Simglucose v0.2.1 (2018) [Online]. Available: `https://github.com/jxx123/simglucose`. Accessed on: 07-24-2021.

**Real-world: HIV Data.** We identified individuals between $18-72$ years of age from the EuResist database (Zazzi et al., 2012) comprising of genotype, phenotype, and clinical information of over $65,000$ individuals in response to antiretroviral therapy administered between $1983-2018$. We focus on a subset of $32,960$ patients' genotype, treatment response, CD4+, and viral load measurements, gender, age, risk group, number of past treatments collected over on average 14 years (aggregated at $4-6$ month intervals). Our action space consists of the 25 most frequently occurring drug combinations, while our state space consists of 100 continuous states of cell counts and viral loads. Since the virus evolves in response to drug pressure, the problem is inherently non-stationary. Our data is collected using a standard first-line therapy provided by clinicians. For our first case study (Case-I), we use a candidate clinician-provided policy as $\pi_{\text{tar}}$. Given data from the first-line therapy, we investigate whether deferring to second-line therapy ($\pi_0$), as proposed by standard medical guidelines in response to potential drug resistance(Saag et al., 2020), improves long-term outcomes. The non-stationary behavior policy is the first line therapy estimated using Q-learning. For our second case study (Case-II), data is collected from a non-stationary behavior policy which corresponds to a first-line therapy typically used for treating patients of subtype C. Using the same candidate $\pi_{\text{tar}}$ as in Case I, we then examine whether deferring to first-line therapy, given by clinical collaborators, for patients of subtype M (due to potential drug resistance) improves long-term outcomes.

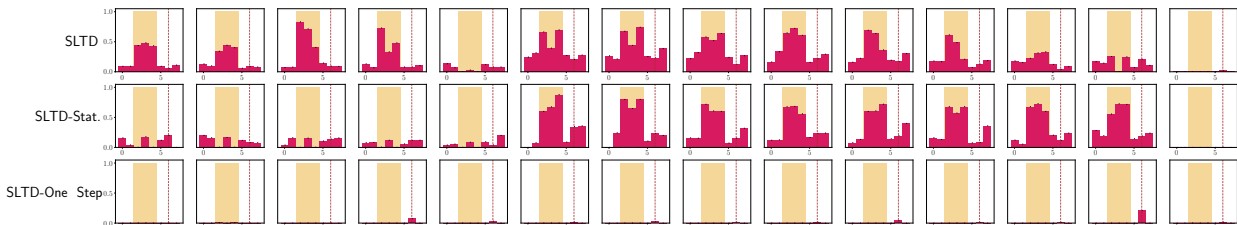

Figure 2: Learned deferral probabilities for SLTD (top row), SLTD stationary (second row), SLTD one-step (third row), and Augmented-MDP (red dotted line) for $c = 0$. Each row is a method (Row 1 is SLTD, Row 2 is SLTD-Stationary, and Row 3 is SLTD-One Step). We are plotting $\tilde{g}_{\pi_{\text{tar}}}(s, t)$ which is a function of states $s$ and time $t$. Since this is a function of two dimensions, we represent the variation along $t$ in a row and then for each fixed $t$, we show the different values this function can take for different states $s$. According to Synthetic data design, target policy always takes suboptimal actions in the yellow region. That is, the optimal deferral policy is to defer for all states $2, 3, 4$ for all $0 \le t \le 15$. That is, $g_{\pi_{\text{tar}}}(s, t) = 1 \forall s \in \{2, 3, 4\}$ and $0 \le t \le 15$ and 0 otherwise. Further, the dynamics change over time so that the optimal action flips, as well as the noise increases when $5 \le t \le 12$ requires deferral more often. Thus, shaded yellow regions are the region of pre-emptive deferral. SLTD, which models non-stationarity defers adaptively and early in the shaded yellow region (top row) and increases the deferral probability when dynamics change. SLTD-stationary does not learn calibrated probabilities in the yellow region over time and only defers when the *average* dynamics of the environment require deferral. SLTD-one-step and Augmented-MDP (dotted red line) only defer in state 6 when the reward is negative and is not pre-emptive.

**Baselines.** We compare to the following baselines.

**Mozannar et. al.** (Mozannar and Sontag, 2020): This is a supervised method using a consistent loss function to learn-to-defer. It learns an augmented regressor to defer or recommend treatment myopically (independently at every time step). When the model defers, the clinician's treatment recommendation is used.

**Madras et. al.** (Madras et al., 2017): This is an alternative supervised learning-to-defer method. This baseline learns separate regressors to defer and recommend treatments. We modify it to use $\pi_{\text{tar}}$ to recommend and learn the rejection function to defer to $\pi_0$. Note that both the supervised learning-to-defer methods are trained to predict action targets.

**Augmented-MDP**: A conceptual contribution of SLTD is to defer by comparing outcomes by delaying deferral with some knowledge of the expert policy. The deferral action itself is considered to augment the MDP action space. We explore a baseline that uses Value Iteration in this augmented MDP. Comparing with this baseline helps evaluate the utility of deferring based on outcomes of delayed versus immediate deferral. This baseline will defer permanently to the expert, and knowledge of an expert policy is not assumed. This

augmented MDP has action-space is $\mathcal{A} \cup \perp$, an augmented state-space $\mathcal{S} \cup s_{defer}$ ($s_{defer}$ is the deferred state), and defers based on the cost $c$. This baseline models non-stationary dynamics, and is designed for sequential settings. However since this method defers permanently to the expert, it incurs a larger deferral cost. In our experiments, all values are plotted *without* the cost to reflect actual environment outcomes.

**SLTD-Stationary**: To assess the impact of misspecifying the non-stationarity, we compare to a variant of SLTD that assumes the dynamics (and rewards) are stationary while allowing the method the flexibility of learning a non-stationary deferral policy.

**SLTD-One Step**: We compare to a myopic version of SLTD that defers based on the immediate reward. The key difference with the myopic Madras et. al., Mozannar et. al. baselines is that SLTD-One Step models the dynamics and the uncertainty on the immediate reward. Thus, this baseline accounts for future deferrals while deferring myopically.

**Ablations for Uncertainty Modeling**: For all SLTD variants, we evaluate the utility of accounting for different sources of uncertainty, more specifically the *modeling uncertainty* to estimate the probability of improving outcomes via delayed deferral. In SLTD, *modeling uncertainty* is accounted for by sampling multiple ($K$) MDPs (Equation 3) from the posterior dynamics distribution, over which our outcomes are averaged. Higher variability across $K$ indicates higher modeling uncertainty. Hence, in Equation 3, using $K = 1$ assumes a perfect estimate of the dynamics model and only accounts for the irreducible stochasticity of the environment. A larger $K$ accounts for potential variability in estimation (original SLTD formulation). If our modeling uncertainty in the environment is indeed large, we anticipate the choice of $K$ to have a larger effect on SLTD's performance. Modeling uncertainty can be large when there is insufficient data to fit the target function class of the dynamics.

## 5.1 Evaluation Metrics.

We use the following evaluation metrics for assessing our proposed models.

**Value Estimation.** Once a deferral policy is learned, evaluating its utility using offline data is the problem of off-policy evaluation (OPE). We use the following estimation procedure for evaluating the Value of a deferral policy:

1. Rollout with true dynamics: For Synthetic data and Diabetes , we have access to the *true dynamics*, although Diabetes  data may be prone to discretization error. Here we can roll out the trajectories under the true dynamics using deferral and collect the true value estimates. In our experiments, we collect 10000 trajectories for all datasets.

2. Rollout with estimated dynamics: For HIV data, we do not have access to the true dynamics. Hence we use the same procedure above but with estimated dynamics, which are estimated using Maximum-Likelihood. This evaluation can be used when there is sufficient domain knowledge to rely on the utility of MLE estimates of dynamics. In our HIV data, the MLE estimates are considered state-of-the-art in clinical knowledge of HIV treatment Parbhoo et al. (2017). However, in general, relying on an MLE estimate is prone to a biased value estimate.

3. Self-normalized Importance Sampling (IS): For real-world data where true dynamics are unavailable and relying on MLE estimates of the dynamics may lead to bias, we provide an IS estimate of the value. This metric will likely be used in most real-world scenarios while choosing the best deferral policy. Note that IS is only asymptotically unbiased and suffers from large variance issues. Hence we use a Self-normalized IS estimate. Note that IS can only be used when the assumption of overlap as well as no unobserved confounding holds.

**Deferral Frequency.** In order to assess the trade-off of deferral frequency and value attained, we use the true dynamics for Synthetic data and Diabetes  data to roll out trajectories and estimate deferral frequency. For HIV data, we use the Maximum-Likelihood dynamics to roll out trajectories to estimate deferral frequency based on the consistency of value estimates of Self-normalized IS and MLE dynamics, which then allows us to use the MLE dynamics to roll out trajectories. Please see further justification of this choice in Section 6.

| Method | Synthetic | | Diabetes | | HIV Case-I | | HIV Case-II | |
|---|---|---|---|---|---|---|---|---|
| | Value estimate using true dynamics (mean ± 2 s.e.) | Self-Normalized IS (mean ± 2 s.e.) | Value estimate using true dynamics (mean ± 2 s.e.) | Self-Normalized IS (mean ± 2 s.e.) | Value estimate using MLE dynamics (mean ± 2 s.e.) | Self-Normalized IS (mean ± 2 s.e.) | Value estimate using MLE dynamics (mean ± 2 s.e.) | Self-Normalized IS (mean ± 2 s.e.) |
| SLTD | $6.837 \pm 0.229$ | $3.710 \pm 0.078$ | $18.882 \pm 0.151$ | $-1.76e+05 \pm 1.00e+04$ | $\mathbf{18.628 \pm 0.073}$ | $16.457 \pm 0.351$ | $\mathbf{25.192 \pm 0.021}$ | $\mathbf{21.695 \pm 0.217}$ |
| SLTD-Stat. | $6.464 \pm 0.043$ | $4.170 \pm 0.202$ | $4.312 \pm 0.371$ | $-2.27e+06 \pm 7.30e+03$ | $17.198 \pm 0.170$ | $\mathbf{17.261 \pm 0.196}$ | $17.859 \pm 0.586$ | $11.281 \pm 0.397$ |
| SLTD-One Step | $6.324 \pm 0.019$ | $-1.283 \pm 0.015$ | $18.853 \pm 0.105$ | $-1.16e+05 \pm 9.97e+03$ | $6.859 \pm 0.027$ | $10.791 \pm 0.331$ | $7.103 \pm 0.019$ | $11.227 \pm 0.374$ |
| SLTD (K=1) | $6.775 \pm 0.397$ | $3.819 \pm 0.125$ | $14.574 \pm 0.931$ | $-1.87e+05 \pm 6.55e+04$ | $8.611 \pm 0.025$ | $11.159 \pm 0.517$ | $22.187 \pm 0.421$ | $17.864 \pm 0.661$ |
| SLTD-Stat. (K=1) | $6.455 \pm 0.069$ | $4.028 \pm 0.265$ | $5.054 \pm 1.984$ | $-2.17e+06 \pm 1.08e+05$ | $6.527 \pm 0.017$ | $8.659 \pm 0.182$ | $18.551 \pm 0.236$ | $17.877 \pm 0.157$ |
| SLTD-One Step (K=1) | $6.332 \pm 0.030$ | $-1.170 \pm 0.081$ | $15.664 \pm 0.839$ | $-9.46e+04 \pm 4.81e+04$ | $6.173 \pm 0.215$ | $6.258 \pm 0.539$ | $7.136 \pm 0.142$ | $4.178 \pm 0.131$ |
| Augmented-MDP | $3.611 \pm 0.028$ | $2.731 \pm 0.000$ | $-0.742 \pm 0.012$ | $-2.64e+06 \pm 0.000$ | N/A | N/A | N/A | N/A |
| Mozannar et. al. | $\mathbf{7.828 \pm 0.163}$ | $4.111 \pm 0.073$ | $-4.253 \pm 10.811000$ | $-2.37e+06 \pm 1.39e+04$ | $6.857 \pm 0.255$ | $2.139 \pm 0.381$ | $8.259 \pm 0.133$ | $3.168 \pm 0.281$ |
| Madras et. al. | $3.318 \pm 0.702$ | $\mathbf{4.743 \pm 0.083}$ | $\mathbf{35.955 \pm 0.488}$ | $-8.148 \pm 0.026$ | $9.238 \pm 0.301$ | $6.829 \pm 0.116$ | $4.831 \pm 0.117$ | $4.173 \pm 0.027$ |
| $\pi_{\mathrm{tar}}$ | $1.329 \pm 0.055$ | $-1.397 \pm 0.000$ | $-0.771 \pm 0.033$ | $-2.44e+06 \pm 0.000$ | $4.122 \pm 0.217$ | $3.159 \pm 0.158$ | $3.186 \pm 0.319$ | $3.199 \pm 0.212$ |
| $\pi_0$ | $6.522 \pm 0.044$ | $11.564 \pm 0.000$ | $35.166 \pm 0.114$ | $-8.085 \pm 0.000$ | $8.815 \pm 0.027$ | $6.158 \pm 0.015$ | $7.857 \pm 0.139$ | $4.844 \pm 0.016$ |

Table 1: Expected Value of SLTD compared with baselines for Synthetic data, Diabetes , and HIV data. i) True dynamics (first column), ii) Self-normalized Importance Sampling estimate (second column). For Synthetic data, IS estimate is biased relative to the true dynamics suggesting IS estimates may not be reliable due to finite samples. The best value according to all methods for Diabetes is Madras et. al. suggesting benefits to myopic deferral for this data. However, the frequency trade-off shown in Table 8 shows that Madras et. al. achieves this value when it defers (always) to the clinician. SLTD is competitive for Synthetic data and outperforms all other methods for HIV data. According to IS estimates, Madras et. al. provides the best value for Synthetic data. For Diabetes , severe overlap issues result in unreliable IS estimates, except for Madras et. al. which matches $\pi_0$ as it completely defers to the clinician. IS-estimates are consistent with MLE-based estimates for HIV where SLTD consistently outperforms all baselines. Figure 3 shows the deferral frequency versus value trade-off for all datasets.

# 6   Results

**Optimizing for long-term outcomes learns qualitatively different deferral policies.**   Our deferral policy is a non-stationary stochastic function $\tilde{g}_{\pi_{\mathrm{tar}}}(s, t)$ which we threshold. Visualizing $\tilde{g}_{\pi_{\mathrm{tar}}}(s, t)$ enables us to understand the utility of various modeling choices of SLTD. Figure 2 shows the histograms of $\tilde{g}_{\pi_{\mathrm{tar}}}$ for SLTD and its Stationary and One-Step variant when the cost $c = 0$ for Synthetic data. Visualizing without deferral cost allows us to see how adaptive SLTD is without a penalty. Each row corresponds to a method; the x-axis corresponds to time over the horizon $T$. Each box in a row corresponds to a single time point. For a fixed $t$, $\tilde{g}_{\pi_{\mathrm{tar}}}$ is a stochastic function of the states, shown as a histogram.

The yellow shaded region indicates the state space where $\pi_{\mathrm{tar}}$ takes unfavorable actions. Over time, the dynamics change so that the favorable action flips $5 \leq t \leq 12$ and the stochasticity in the dynamics increases requiring more frequent deferrals. Deferring in the yellow region is desirable to pre-emptively avoid landing in a state of 6. SLTD is highly adaptive, and pre-emptively defers in the yellow region. As the stochasticity increases, the probability of deferrals appropriately increases. The stationary variant significantly underestimates the need to defer in states $2, 3, 4$ when $t < 5$. It is only able to pre-emptively defer in regions where the average stochasticity of the estimated dynamics aligns with the environment. The One-Step variant defers only in state 6 and is therefore not preemptive. Augmented-MDP (red vertical line) deterministically defers in state 6. Thus deferring based on the probability of improved outcomes of immediate and delayed deferrals is desirable over alternatives (see also Figures 4, 5, and 6 in Appendix for Diabetes  and HIV data).

It is important to note that SLTD lowers deferral frequency for $t \geq 12$. The reason for this is two-fold. First, it is an artifact of the synthetic data that we use to evaluate SLTD. More specifically, there is an additional form of non-stationarity that kicks in at $t \geq 12$ where we stop adding noise to our dynamics, which resets the dynamics to where they were at $t < 5$. As a result, the deferral frequency is expected to be lower than at $5 \leq t \leq 12$. Second, since SLTD is designed in an offline learning environment, the support of the data matters (for any offline learning algorithm). For $t > 12$, the offline data collected does not have significant support over the "deferral" states $[2, 3, 4]$, which induces an estimation issue. While we expect this to be compensated by uncertainty modeling that we incorporate, lack of data support is a much more fundamental problem in offline learning, and we anticipate requiring better prior knowledge of the dynamics to completely

overcome the bias that is introduced in the deferral frequency. This bias further lowers deferral frequencies even more compared to $t < 5$. Nonetheless, the learned deferral policy improves as can be seen from Table 1 and noting that the value estimate we have from the optimal policy for Synthetic data was estimated to be $\approx 8$. Mozannar et. al. is able to achieve close to optimal by deferring over 80% of the time.

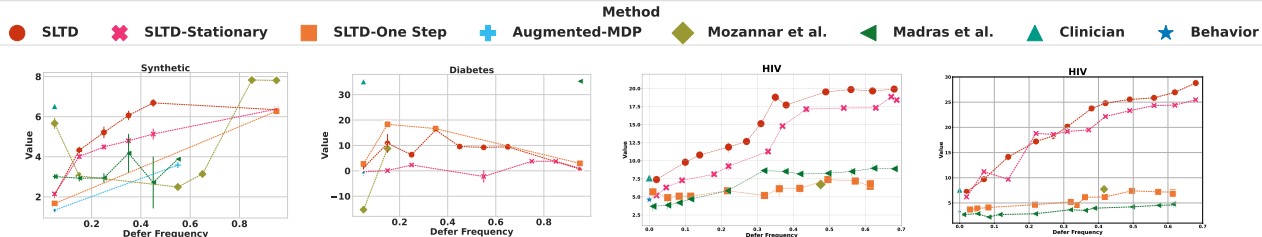

Figure 3: The trade-off of deferral frequency and value attained. Synthetic data and Diabetes use true dynamics, while HIV uses Maximum-Likelihood estimates for value estimates. The plot shows the expected value (higher is better) for a sweep over deferral cost (corresponding to best performing hyperparameters for each baseline) to the deferral frequency (lower is better). SLTD achieves the best trade-off and better value by deferring pre-emptively. The stationary variant is unable to attain high values for the same deferral frequency. The One-Step variant cannot improve over $\pi_{\text{tar}}$ in MDPs where the effect of interventions is not myopic (Synthetic and HIV) and achieves good performance by modeling the dynamics in Diabetes, compared to Mozannar et. al.. Madras et. al. baseline performs well on Diabetes data suggesting this data is myopic, though non-stationary (considering deterioration of SLTD-Stationary, and the high deferral frequency of Madras et. al. baseline. Augmented-MDP is unable to achieve good performance indicating the benefits of deferring by explicitly quantifying the impact of delaying deferral as in SLTD over Augmented-MDP's Value Iteration method. SLTD and the stationary variant achieve the best trade-offs in HIV data. Deferral trade-offs when relying on Self-normalized IS for model-selection are provided in Tables 8 and 11 in the Appendix.

**SLTD improves long-term outcomes.** Table 1 shows the value (higher is better) for all baselines using the different value estimation methods outlined above. We have shown the highest value for each value estimate independently. For HIV data, we do not see significant differences between MLE dynamics and IS evaluation metrics. SLTD unanimously outperforms all baselines. Table 1 shows significant biases in the IS estimate compared to the true dynamics for Synthetic data data. Based on IS estimates, Madras et. al. performs the best for Synthetic data. Table 8 in Appendix shows the frequency of deferrals and the corresponding cost hyperparameter, if IS value estimates are used to select the best deferral methods. This performance is obtained when there is no cost to deferring ($c = 0$) resulting in deferral frequency of over 50%. For $c > 0$, we see a drop in value to less than 2 for Madras et. al. (see Figure 3). This suggests that SLTD is still efficient and able to reach a high value for fewer deferrals in Synthetic data data. For Diabetes data, Madras et. al. performs the best according to both Self-normalized IS and using the true dynamics. However, Figure 3 suggests that the Madras et. al. baseline always defers to clinician and never has a deferral frequency less than 1. Due to significant overlap issues, IS-estimates for Diabetes are unreliable except for Madras et. al. which matches clinician policy (due to permanent deferral). As such there are untestable overlap issues using IS that need further attention in such non-stationary environments.

Based on the true dynamics, Augmented-MDP baseline is not preemptive despite modeling the non-stationary dynamics. Hence, deferring by comparing outcomes of delayed deferral is a better alternative, specifically Value Iteration used in Augmented-MDP. Additional benefits of modeling the dynamics are clear from the improved performance of all SLTD variants compared to the myopic Mozannar et. al. baseline when $c > 0$. Mis-specification of dynamics (SLTD-Stat.) results in worse performance. SLTD-Stat. defers more often to achieve comparable performance. Figure 3 demonstrates this trade-off is general, for all choices of deferral costs and other parameters. In Figure 3, the x-axis corresponds to deferral frequency (lower is better) and y-axis the value attained (higher is better). For Synthetic data and Diabetes , value estimates using the true dynamics are shown. For HIV data, since we do not have access to the true dynamics, the deferral frequency is estimated using an MLE estimate of the dynamics. From Table 1, we can see that all value estimation

procedurs provide comparable estimates for HIV. SLTD achieves the best trade-off for Synthetic data and HIV.

Further, pre-emptive deferral allows SLTD to reach close to optimal (value average obtained is $\approx 7.0$) in Synthetic data, though Mozannar et. al. achieves a higher value when deferral cost is 0. SLTD-One Step only relies on immediate rewards failing to improve long-term outcomes for HIV. However, as long as we model the dynamics appropriately, even myopic deferral using SLTD-One Step is beneficial for Diabetes compared to Mozannar et. al.. This is possible when the effect of interventions is observed myopically, as is the case in Diabetes data since modeling the dynamics and impact of future deferral is beneficial to characterize. Madras et. al. performs well on Diabetes data suggesting optimal actions don't significantly deviate in target data and that its design of training a rejection function worked better than the loss function design of Mozannar et. al.

| | Defer Time $t_d$ | Total Uncertainty | Modeling Uncertainty | Mean Outcome |
|---|---|---|---|---|
| Synthetic data | 3 | 26.190 | 0.233 | 3.42 |
| Diabetes | 3 | 3418.17 | 34.160 | 73.669 |

Table 2: Interpreting first time of deferral for a sample trajectory. Modeling uncertainty remains low in all cases whereas in comparison, total variance is high. This indicates irreducible stochasticity of the dynamics is the primary source of uncertainty. Additional results are in Appendix.

**Ablations for uncertainty modeling.** We study the utility of accounting for modeling uncertainty in our framework. As described in Section 4, multiple sources of propagated uncertainty contribute to variability in estimated outcomes. Modeling uncertainty is crucial to account for in a model-based framework. Here we evaluate the impact of not accounting for this uncertainty on SLTD's performance.

If modeling uncertainty is high, variability of the sampled MDPs used to estimate Equation 3 will be higher. Evaluating for $K = 1$, will evaluate the impact of ignoring this uncertainty. In Table 1 (see also Figure 8 in Appendix), we demonstrate the results with $K = 1$ for all SLTD variants. We do not observe significant differences for Synthetic data and Diabetes indicating that our modeling uncertainty is low in these data. The difference is higher in HIV suggesting the importance of accounting for this uncertainty for real-world HIV data. Such analysis is crucial to understanding whether our modeling assumptions are reasonable.

**Decomposing uncertainty in SLTD can help interpret deferral.** Conveying the type of uncertainty to a domain expert can help identify the dominant source of uncertainty that resulted in a deferral to their standard practice (expert policy). Table 2 shows this decomposition for one timepoint for discrete data. In each case, the modeling uncertainty is a small fraction of the total uncertainty. This suggests that systematic non-stationarity is the dominant source of uncertainty which generally cannot be reduced by collecting data and may require careful interventions beyond the standard policy. Knowledge of the amount of model uncertainty can enable users to further improve decision-making through data collection or improving model assumptions.

## 7 Discussion

We proposed SLTD, a learning-to-defer framework for sequential settings using offline model-based RL. We learn a deferral policy by quantifying the impact of delaying deferral to the future. SLTD can defer based on long-term outcomes and learns a pre-emptive deferral policy. Further, we emphasize a model-based RL method that captures the dynamics of the environment, particularly non-stationarity. Modeling the non-stationarity of the environment allows deferring adaptively. Misspecifying non-stationarity leads to significantly more deferrals to improve long-term outcomes. We demonstrate that existing learning-to-defer frameworks are myopic. That is, these methods do not learn a pre-emptive policy even in sequential settings as they focus on the immediate consequences of actions. We further demonstrate the utility of accounting for all potential sources of stochasticity to quantify the impact of delayed deferral. Explicit characterization of the probability of improving outcomes is beneficial to prevent over-estimation of the benefits of delaying deferral. We further interpret deferral decisions of SLTD by decomposing the *long-term* propagated uncertainty.

**Limitations and Future Work.** While quantifying the uncertainty is useful, especially to compensate for the fundamental challenge of data support in offline reinforcement learning, modeling uncertainty through non-stationarity is costly. Developing a model-free framework is an important aspect of future work. Uncertainty quantification may not be able to compensate for cases of severe data support issues, which may result in biases in the learned SLTD policy. In this case, better prior knowledge of the dynamics is necessary. SLTD assumes no hidden/unobserved confounding by relying on an MDP data-generating assumption. Thus our current uncertainty quantification does not account for the added epistemic uncertainty due to potential hidden confounding. Incorporating clinically motivated sensitivity models to account for unobserved confounding is an active area of our future work. SLTD can account for some deviations from the expert policy, as well as the potential for its recommendations to be overridden, though significant deviations should be modeled as an online human-in-the-loop system. In this case, SLTD can serve as a reliable warm-start policy that could be further improved using (online) human input. We also posit that our current $\epsilon$-greedy version will prove to be more conservative compared to such an online framework, as we expect real human decisions to be more informed (modulo their own biases) than a noise model that is $\epsilon$-greedy. Nonetheless, rigorously testing this human-in-the-loop learning-to-defer framework is left to future work. Finally, note that if the type of non-stationarity we would like to defer against is not observed in the offline data, the deferral policy may over/understimate the need to defer.

**Ethical considerations.** SLTD is a technical proof-of-concept to defer to an expert by accounting for long-term effects, assuming that the expert is better at increasing value over the current policy in certain regions. In practice, an expert policy may not be bias free. Thus, deferral may result in biased decisions if the expert is biased. Such bias may be exacerbated due to potential sources of hidden confounding (Gottesman et al., 2018). While we are not focused on addressing bias, exposing uncertainties may encourage expert introspection. Nonetheless, deferring is better when an automated decision may be harmful.

# 8 Acknowledgements

This material is based upon work supported by the Center for Research on Computation and Society (CRCS) at Harvard University, the National Science Foundation under Grant No. IIS-2007076. Any opinions, findings, and conclusions or recommendations expressed in this material are those of the author(s) and do not necessarily reflect the views of the funding agencies.

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
