# OpenReview forum: "Learning-to-defer for sequential medical decision-making under uncertainty"
_TMLR — Accepted by TMLR_

### Review · Reviewer_yj16 · 2022-12-31

**Summary Of Contributions:**


This paper aims at learning a policy in sequential medical decision-making where the system may decide to defer to a domain expert in case the outcome is expected to be worse than deferring. Furthermore, the paper focuses on non-stationary situations where uncertainty is particularly high, and where deferring may be especially relevant. Finally, the proposed methods are evaluated on synthetic, semi-synthetic, and real-world data, in order to validate their effectiveness.


**Audience:**

Yes

**Broader Impact Concerns:**

- The model is motivated for a health care setting. It is hard to see how it is possible to quantify the value of the deferred policy. I can see that it is not an issue in the experiments in this paper. But I have doubts about its practical usefulness. See questions to the authors above.


**Claims And Evidence:**

No

**Requested Changes:**

- It looks to me that the contribution of this work is more on the algorithmic side since there lack justifications or supporting details for the clinical practice. On the methodology side, there are lot of clarity issues regarding problem setups/ technical details. For evaluation, the sample efficiency should be reported as well. On the clinical side, I am not completely sure about its broad applicability. Although it is valuable, it could discuss more about its assumptions, limitations, and ethical concerns.
- See my comments in the main weaknesses above.
- How to select the conjugate priors?
- What's the difference between long-term outcomes and short-term outcomes in health setting in this paper?

**Strengths And Weaknesses:**

Strengths:
- It is an interesting setup and somewhat novel: This work advances learn-to-defer algorithms by considering them in non-stationary, sequential settings and by identifying the value of delaying deferral, although some of its ideas overlap with that of safety and robustness in reinforcement learning (RL).
- Resources and Reproducibility: Simulation data are open-source and code was made available.

Weaknesses:
- The presentation of the paper could be improved and sometimes it's hard to follow. For example, it claims that this work focuses on offline model-based RL. According to the paper, the target policy is learned from the offline data whereas the behavior policy is provided by the domain expert. What are the differences between target policy and behavior policy? How is this behavior policy provided, i.e., online or offline? Is the target policy expected to learn or to evaluate? I am confused since it looks to me that this work should concern more about the setting of deploying a learned offline policy into an online changing environment. But if it's a totally offline setting, I am wondering how does dynamics change. In figure 1, it is unclear what are those symbols, i.e., pills, syringes. Are they available actions at each time point? What are the arrows with different colors? What is the expectation of outcomes here? When dynamics are changing, how's the optimal policies for them? Will the optimal policies change as well or they are the same? Which one is SLTD? I didn't see the green arrow. In figure 2, what does each plot in the row mean? I don't understand what is the difference. What's y-axis? It looks like it should be the value of \hat{g}_{{\pi}_{tar}}. But there is no value of 1 (defer) in those plots, how does this \hat{g}_{{\pi}_{tar}} do? What's the optimal policy or ground truth here?
- Since health care setting is quite different from the general setting, it's necessary to clarify which kind of medical decision-making task this work focuses on? Besides, the setups, assumptions, and patient outcomes could be different a lot under different health care situations, e.g., critical care, chronic care, etc. What are they in this work? How to define the state space, action space, and reward function under this health care setting? How to obtain the target policy and behavior policy in each dataset?
- Since this paper claims its applicability in a data-scarce setting, it would be more convincing if the number of data samples used for training can be provided. This is important, particularly in the health care setting.
- Why the non-stationary dynamics are represented by a sequence of MDP? What are the differences among those MDPs? Do they share the same optimal policy?
- What's the ground truth for the value of the deferred policy?


Minors:
- Typo "the target policy $\pi_0$ may not uniformly improve over $\pi_{tar}$ for all states"
- Typo “such that such that…”.

---

### Review · Reviewer_w4hA · 2023-02-01

**Summary Of Contributions:**

In this work, the authors introduce a new framework to better model decision making problems involving deferrals to a human expert. This framework, called Sequential Learning to Defer (SLTD) treats interventions as a multi-step decision making process, and aims to model long-term consequences of interventions. To this end, they introduce a model-based method that factors in not only long-term outcomes, but also non-stationarity that might exist in the dynamics of the environments. Fundamentally, they learn a 'deferral function' that estimates the probability that a costly 'expert' policy $\pi_0$ (i.e., a human) improves over an alternative 'target' policy $\pi_{\text{tar}}$ (i.e., an automated system).  This deferral function itself constitutes a policy that determines whether, for a given timestep, deferral to a human expert should take place. By accounting for future uncertainty, they are able to defer pre-emptively, and improve overall outcomes. Moreover, by taking a fully Bayesian approach, they are able to model key sources of uncertainty, such as dynamics, which allows them to handle non-stationarities that may cause maximum-likelihood approaches to be over-confident/miscalibrated in their estimations.

They show that their approach to their introduced problem setting performs significantly better than previous myopic approaches (that don't take into account future outcomes), and perform a set of ablations to show that key modelling decisions, such as modelling future values, are key to strong performance.

**Audience:**

Yes

**Broader Impact Concerns:**

None, I believe the authors list these in the paper already.

**Claims And Evidence:**

Yes

**Requested Changes:**

I would appreciate clarity on the points raised above in the weaknesses (and potential rewriting to make it clearer in the manuscript), and will base my recommendation on the answers.

**Strengths And Weaknesses:**

Strengths:
* The problem setting seems well motivated, and is a nice extension to existing deferral-based frameworks
* The proposed method appears to account for uncertainty in the deferral process, and significantly outperforms myopic approaches

Weaknesses:
* I found a few parts difficult to understand, and would appreciate clarity:
    * Figure 2 is unclear to me; while SLTD appears to learn to defer earlier than SLTD-Stat, it appears to stop deferring at t=13? Is this result expected? My understanding is we should still defer in states 2,3,4 after t=12 as the target policy still tries to pick sub-optimal actions
    * In the diabetes data, non-stationarity is created by swapping the adolescent patient property with an alternative pateint over time. Is this intended to model a realistic scenario of how patient properties may change? Wouldn't a different process be more realistic (e.g., a slowly increasing/decreasing value, such as body weight)? I ask as this design choice seems to significantly favour approaches that can model sudden changes in dynamics.
    * Does SLTD require knowledge a-priori of what an expert will perform? From my understanding, to calculate the expected return under the mixture policy, one must have access to the decision that the expert may perform in the future? Is this a realistic assumption given an actual expert could be a human? I believe the expert policies considered here are still synthetic.

---

> ### Author Response · Authors · 2023-02-16
> **Thank you for your comments. We have addressed your concerns and updated the manuscript to reflect all changes**
>
> > Figure 2 is unclear to me; while SLTD appears to learn to defer earlier than SLTD-Stat, it appears to stop deferring at t=13? Is this result expected? My understanding is we should still defer in states 2,3,4 after t=12 as the target policy still tries to pick sub-optimal actions.
>
> This is an excellent question. The reason why SLTD lowers deferral frequency for $t\geq 12$ is two fold. First, it is an artifact of the synthetic data that we use to evaluate SLTD. More specifically, there is an additional form of non-stationarity that kicks in at $t\geq 12$ where we stop adding noise to our dynamics, which resets the dynamics to where they were at $t<5$. As a result, the deferral frequency is expected to be lower. We have updated the description in the text to make this more explicit now. Second, since SLTD is designed in an offline learning environment, the support of the data matters (this would be the case for any offline learning algorithm). In later times, the offline data collected does not have significant support over the “deferral” states [2,3,4], which induces an estimation issue. While we expect some of this to be compensated by uncertainty modeling that we incorporate, lack of data support is a much more fundamental problem in offline learning, and we anticipate requiring better prior knowledge on the dynamics to completely overcome the slight bias that is introduced in the deferral frequency, which further lowers deferral frequencies even more compared to $t<5$. We have added this explanation to the manuscript, and also made this data support challenge explicit in our overall discussion of limitations. We visualized the data support and added it for your reference here: https://osf.io/hfteu/?view_only=442cc736804844b1a7443aa6ca3e1435. Thank you for the insightful question!
>
> > In the diabetes data, non-stationarity is created by swapping the adolescent patient property with an alternative patient [sic] over time. Is this intended to model a realistic scenario of how patient properties may change? Wouldn't a different process be more realistic (e.g., a slowly increasing/decreasing value, such as body weight)? I ask as this design choice seems to significantly favour approaches that can model sudden changes in dynamics.
>
> Under the hood, changing the patient properties to an alternative one is changing the stomach solids, liquids, glucose production, renal secretion, and glucose and insulin kinetics slowly over time, which is a proxy to test the utility of our approach. We are indeed increasing/decreasing these values slowly (by changing the weighting of the parameters smoothly over the horizon). Introducing non-stationarity in more realistic via obvious changes like weight has proven challenging for us in the simulator. Nonetheless these are not sudden changes since the weighting varies very smoothly over the horizon. Figure 4 in the Appendix demonstrates the resulting deferral policy. We can see a smooth transition in how deferral probabilities increase over time until they stabilize much later in the horizon. This suggests that the only key to success is that the offline data we have access to should reflect the type of non-stationarity we anticipate in the real world. If changes are smoother than our horizon, no method will be able to address this issue. We have added further discussion on this in our Diabetes experiments and Discussion section.
>
> > Does SLTD require knowledge a-priori of what an expert will perform? From my understanding, to calculate the expected return under the mixture policy, one must have access to the decision that the expert may perform in the future? Is this a realistic assumption given an actual expert could be a human? I believe the expert policies considered here are still synthetic.
>
> SLTD requires that the expert policy we defer to is known. It does not require perfect knowledge of what actions an expert will perform in the future.  Since $\pi_0$ can be stochastic, we of course can deal with uncertain actions. Thus, we do not make explicit assumptions on the specific actions an expert will take. SLTD accounts for the added stochasticity by explicitly taking expectations over actions $a \sim \pi_0$, when we estimate the value of deferral to estimate our deferral policy. The expert policy is a reflection of common clinical practice in our target domain of healthcare. This is not unrealistic as a lot of clinical knowledge is aggregated into clinical protocols a human expert, such as a clinician, would use in practice. In addition, to further reflect realistic scenarios, we use an $\epsilon$-greedy version of this policy in our experiments, which will allow for further deviations from the expert policy, making it a better proxy for realistic settings. Note further that we only use this strategy in our experiments and do not assume an $\epsilon$-greedy policy class, so as to not restrict the utility of SLTD.
>
> [Contd.]

---

### Review · Reviewer_nvhL · 2023-02-06

**Summary Of Contributions:**

The authors present a framework for learning to defer sequential medical decisions based on Bayesian model-based reinforcement learning in Markov decision processes. The problem is posed as learning a policy that chooses adaptively between executing a given target policy and deferring to an expert. The policy is learned by optimizing the cumulative reward given every time the target policy outperforms the behavior policy by a certain margin, as estimated by a learned model. This approach is evaluated in a series of experiments based on simulated and real-world data.

Claimed contributions:
- Defining learning-to-defer problem for sequential settings as offline model-based reinforcement learning (RL)
- Show that modeling non-stationarity in environments provides a better trade-off of improving outcomes versus the frequency of deferrals
- Demonstrate that deferral methods that defer myopically, based on the immediate outcome, still benefit from modeling system dynamics
- Justify how the decomposition can guide experts to potential interventions
- The work is applicable in many safety-focused, data-scarce, non-stationary settings where online policy improvement is not allowed due to ethical or practical constraints

**Audience:**

Yes

**Broader Impact Concerns:**

- Ignoring the impact of selection bias in offline evaluation of clinical decision making policies is dangerous and must be avoided. See e.g., "Guidelines for reinforcement learning in healthcare" by Gottesman et al.


**Claims And Evidence:**

No

**Requested Changes:**

- The paper would benefit from taking its motivation from a real-world clinical task, instead of a general clinical environment. This would add credibility to the framing of learning-to-defer, and perhaps change the setting slightly.

- The work should be better contextualised in the literature of off-policy evaluation and causal estimation.

- The experiments should add missing details on comparisons and evaluation metrics (there is plenty of space until the 12-page limit is reached).

- Typo in Section 3: "In practice, the target policy π0 may not uniformly improve over πtar for all states." It should be the reverse, no?

**Strengths And Weaknesses:**

## Strengths:

- Research on making the deployment of new clinical policy more practical is important, and the learning-to-defer framework fits broadly with this intention (although the precise implementation of it may need more work).

- The contributions are mostly well-presented and easy to follow.

## Weaknesses:

- The intended application is not convincing to me, since it forces a clinician to alternate between giving advice to a machine and letting the machine decide what to do. This seems infeasible to implement in practice, which is the main motivation behind this work. In case the clinician can override the machine's decision (in case it didn't defer), the machine's model is incorrect (since this action is not part of the model) and the value of the future mixed policy is inaccurate. This is a big difference between the present work and the baseline models used for comparison: a) these aim only to do prediction, not decision-making and b) once the prediction is made, the expert/user can do whatever they want with it.

- The paper mostly ignores the huge body of work on (off-policy) policy evaluation (OPPE), see for example "A Review of Off-Policy Evaluation in Reinforcement Learning" (Uehara et al., 2022). This is unfortunate since the present method requires solving the OPPE problem in Algorithm 1, or alternatively, the off-policy policy optimization problem. In particular, success of Algorithm 1 relies on a highly accurate model M of system dynamics and responses to decisions. Finding such a model is no trivial feat and relies on strong assumptions such as causal sufficiency/exchangeability/ignorability, common support for actions, etc. In fact, the topic of causality is completely absent in the present work, which renders the method invalid unless additional assumptions (such as the above) are made. Ways of recovering from violations of such assumptions in the context of policy improvement have already been studied, see e.g., "Confounding-Robust Policy Improvement" (Kallus & Zhou, 2018). Bayesian modelling does not remove selection bias due to unobserved confounding and any measures of model uncertainty should take that into account.

- The paper claims to be well suited for "data-scarce" applications, but given the variance issues of OPPE, I am not convinced by this claim.

- There are many experimental details missing in the main paper. For example,
  - In the simulator examples, how many samples used to fit the transition model in the synthetic data? Is it enough to fit the model M with essentially no error?
  - In the HIV data, the paper does not divulge how the value was computed. In the appendix, it says that it was based on MLE estimates of the system dynamics. Clearly, this biases the value in favour of other model-based estimators of the value. Moreover, this evaluation approach assumes that a perfect model can be found---which renders the present paper superfluous. In other words: this method of evaluation is circular---even computing the metric requires solving the problem the metric is meant to evaluate solutions for.
  - It is unclear how the baselines for Mozannar and Madras are used since these are not developed for decision-making, but for prediction. Are they used to predict the reward? The optimal action?

- Algorithm 1 is only given for the discrete case and the appendix contains no algorithm for the continuous case. As such, it is unclear how the dynamic programming step is performed for continuous states.

- It is not clear if the policy \pi_0 is known or estimated. If it is estimated, it is very unlikely that the estimate is a perfect model of clinical decision-makers since these will have access to more information in any real-world application.

---

### Decision · Action_Editors · 2023-03-17

**Recommendation:** Accept as is

**Comment:**

This is a borderline paper, and the consensus leaned more towards rejection due to the limitation in applicability. However, I consider the TMLR criteria to be satisfied in the sense that the paper is sound, and that there could be interesting follow-up work.

**Audience:**

The applicability of this work, as well as whether it could be built upon, has been questioned by all reviewers. The authors have suggested future work that could be of interest. One reviewer (w4hA) also suggested that this work could be relevant in the context of a hierarchical framework.

[1] On Pathologies in KL-Regularized Reinforcement Learning from Expert Demonstrations, Rudner et al., NeurIPS21

**Claims And Evidence:**

The authors have addressed most clarifying questions, and modified the text. The paper addresses a clear problem, states its assumptions and limitations. No major error was identified.

An important remaining issue was that of applicability, especially as an assumption is that $\pi_0$ is given. The reviewers understand that healthcare has 'protocols' in place in some cases, but these protocols do not simulate a human action, in its complexity and variability. I however appreciated the discussion, and especially the mention of future work with humans in the loop.